# Bridging Fourier and Spatial-Spectral Domains for Hyperspectral Image Denoising

## ABSTRACT

Remarkable progresses have been made in hyperspectral image (HSI) denoising. However, the majority of existing methods are predominantly confined to the spatial-spectral domain, overlooking the untapped potential inherent in the Fourier domain. This paper presents a novel approach to address HSI denoising by bridging the information from the Fourier and spatial-spectral domains. Our method highlights key insights into the Fourier properties within spatial and spectral domains through the Fourier transform. Specifically, we note that the amplitude predominantly encodes noise and photon reflection characteristics, while the phase holds structural information. Additionally, the Fourier transform offers a receptive field that spans the entire image, enabling effective global noise distribution capture. These insights unveil new perspectives on the physical properties of HSIs, motivating us to leverage complementary information exchange between Fourier and spatial-spectral domains. To this end, we introduce the Fourier-prior Integration Denoising Network (FIDNet), a potent yet straightforward approach that utilizes Fourier insights to synergistically interact with spatial-spectral domains for superior HSI denoising. In FIDNet, we independently extract spatial and Fourier features through dual branches and merge these representations to enhance spectral evolution modeling through the inherent structure consistency constraints and continuing reflection variation revealed in Fourier prior. Our proposed method demonstrates robust generalization across synthetic and real-world benchmark datasets, outperforming state-of-the-art methods in both quantitative quality and visual results.

## KEYWORDS

Hyperspectral denoising, Fourier-prior, Spatial-spectral modeling

## 1 INTRODUCTION

Hyperspectral images (HSIs), characterized by numerous contiguous spectral bands, offer rich spectral information with diverse applications ranging from medical diagnosis [5] to geological analysis [26] and vegetation monitoring [25]. However, practical HSIs often suffer from contamination and degradation due to sensor limitations, atmospheric interference, and environmental factors. These factors adversely impact downstream computer vision tasks, particularly in classification [41] and object detection [35]. Consequently, restoring

**Unpublished working draft. Not for distribution.**

*ACM MM, 2024, Melbourne, Australia*
© 2024 Copyright held by the owner/author(s). Publication rights licensed to ACM.
ACM ISBN 978-x-xxxx-xxxx-x/YY/MM
https://doi.org/10.1145/nnnnnnn.nnnnnnn

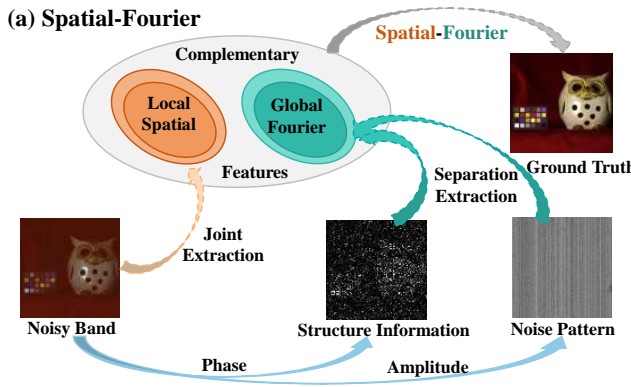

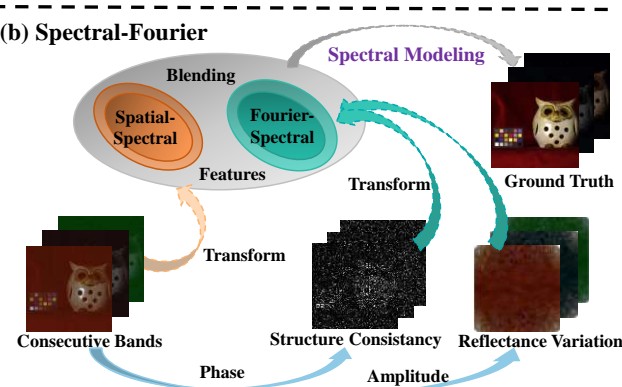

**Figure 1: (a)** In the spatial domain, the amplitude and phase components from the Fourier domain inherently embody global noise patterns and spatial structures, enhancing local spatial representations and addressing diverse data distributions. These properties provide potential solutions to separate noise and structure, boosting model generalization. **(b)** In the spectral domain, the amplitude implies photon reflection variation between different bands, supporting model to reduce spectral distortion, while phase enforces structural consistency, aiding structural recovery.

clean HSIs from noisy observations has become a pivotal challenge in the realms of computational photography and computer vision.

Recent advancements in deep learning (DL) have yielded convolutional neural network (CNN) and Transformer-based methods [2, 16, 17, 20, 30] for end-to-end translation from noisy to clean HSIs. Despite their progress, these methods retain intrinsic limitations. CNN-based approaches extract local features with convolution filters, failing to model long-range spatial dependencies and global noise distributions. Transformer-based methods, although capturing global dependencies and pixel correlations, are computationally intensive and inflexible across various datasets. Importantly,

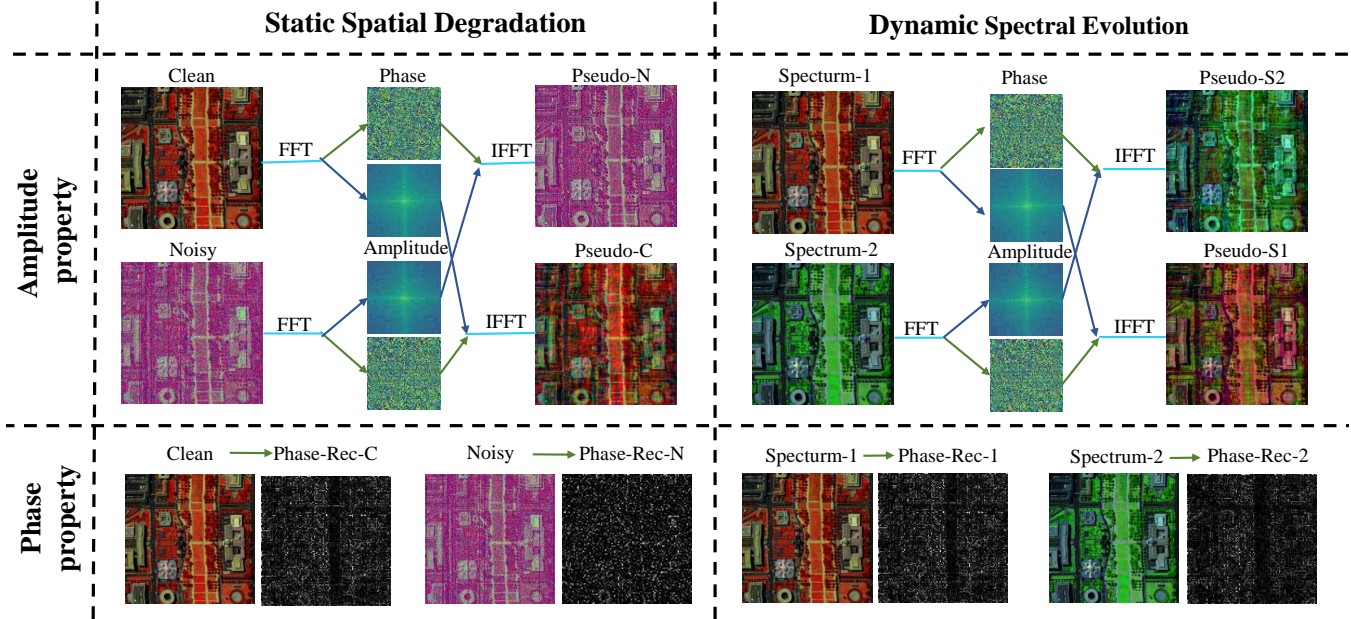

**Figure 2: Examples of physical properties of amplitude and phase components in spatial-spectral domains. For amplitude components, we swap the amplitude on the static spatial degradation and dynamic spectral evolution scenarios. The reconstructed results imply that the amplitude contains most noise information and crucial photon reflection information. For phase components, the similar reconstructed results only using phase representations on the static spatial degradation and dynamic spectral evolution scenarios reveal that the phase components are more related to structure and edge information.**

these DL methods often overlook the intrinsic physical properties of noise patterns and structural information, potentially leading to overfitting on specific synthetic datasets.

To address these challenges, this paper delves into the integration of Fourier knowledge into spatial-spectral domains for HSI denoising. The Fourier priors encompass both spatial and spectral aspects, as depicted in Fig. 1. Specifically, we uncover and leverage the physical properties of amplitude and phase in the Fourier domain. They provide global noise distribution and structural information in the spatial domain, implicitly embed reflection variation information, and enforce structural consistency in the spectral domain. This robust Fourier prior presents a potential solution for enhancing model performance and generalization capabilities. To visualize these above properties, we utilize fast Fourier transforms (FFT) to decompose HSIs into amplitude and phase components following inspiration from [34]. Our investigations unveil intriguing phenomena related to these components, as shown in Fig. 2. For instance, the upper part of Fig. 2 reveals the physical attributes of amplitude. Swapping the amplitude components within the same spatial context yields pairs (**Noisy** & **Pseudo-N** and **Clean** & **Pseudo-C**) with analogous spatial appearances. Similarly, swapping amplitude components of different spectral bands results in spectral similarity (**Specturm-1** & **Pseudo-S1** and **Specturm-2** & **Pseudo-S2**). This signifies that noise and photon reflection data primarily reside in amplitude components. Moreover, in the lower part of Fig. 2, we unveil the physical properties of the phase component. As can be seen, results (**Pha-Rec**) reconstructed only through phase in spatial and spectral contexts encapsulate structural information. Furthermore, Fourier features'

image-size receptive field facilitates global noise distribution and structure information capture.

These insights motivate us to incorporate the complementary information from Fourier domain, enabling the extraction of physical properties that cannot be separated in the spatial-spectral domain. Hence, we introduce the Fourier-prior Integration Denoising Network (FIDNet), an elegant structure designed for effective HSI denoising, as depicted in Fig. 3. Considering the notable modality disparities between Fourier and spatial-spectral domains, effective feature extraction and fusion strategies are essential. FIDNet achieves this by permuting input spectral dimension into batch dimension, enabling focused feature extraction without premature spectral inclusion. Spatial and Fourier domain features are then separately extracted in two branches to overcome inter-domain disparities. The fused features, enriched with Fourier-specific attributes, prove more suitable for spectral evolution modeling by simply detaching spectral dimension back to their original positions, in line with Fig. 1 insights. We simply employ a vanilla 3-D UNet [31] structure for spectral modeling. More sophisticated architectures may indeed yield larger performance gains, but in this paper, such advancements are not the primary focus. Ultimately, the denoised HSI is generated from the independent spatial and Fourier-integrated spectral feature with several 2-D convolution layers. Notably, the proposed FIDNet can handle input HSIs with varying spectral numbers flexibly, avoiding dataset-specific training.

Overall, our work makes following main contributions:

- We reveal the physical properties of amplitude and phase components in the spatial degradation and spectral evolution contexts, and integrate the Fourier and spatial-spectral information for HSI denoising.
- we introduce a novel Fourier-prior Integration Denoising Network (FIDNet), which leverages complementary information from both Fourier and spatial-spectral domains to generate noise-free HSIs. To the best of our knowledge, we are the first to introduce amplitude and phase components for the HSI denoising task, marking a novel advancement in the field.
- We tailor the amplitude-phase block and Fourier-spatial fusion module to exploit the characteristics of amplitude-phase components and effectively capture the complementary relationships from Fourier domain.
- Our FIDNet is simple yet powerful. Experimental results demonstrate that our method surpasses state-of-the-art approaches on synthetic datasets and showcases exceptional generalizability to various datasets.

## 2 RELATED WORKS

### 2.1 Hyperspectral Image Denoising

Mainstream HSI denoising methods can be broadly categorized into two main groups: traditional model-based methods and DL-based methods. Traditional model-based methods [10, 18] rely on hand-crafted mathematical and statistical priors to remove noise from hyperspectral images. For example, Wang *et al.* [28] introduce a tensor-based HSI noise removal algorithm, while He *et al.* [11] devised a low-rank matrix recovery approach with a global spatial-spectral total variation constraint. DL-based methods [33, 38] utilize neural networks to automatically learn and extract features from hyperspectral images for denoising. Wei *et al.* [30] propose a powerful 3D Quasi-Recurrent Neural Network (QRNN) module for effectively capturing spatial and spectral dependencies. Cao *et al.* [2] introduce a sophisticated deep spatial-spectral global reasoning network that effectively combines local and global information for HSI denoising. Li *et al.* [16, 17] introduce vision transformer into HSI denoising task to explore the intrinsic similarity characteristics in both spatial and spectral dimensions. However, these methods rarely explore the potential solutions in the Fourier domain, which is a critical aspect of noise removal.

### 2.2 Fourier Transform in Deep Learning

The Fourier transform is a fundamental tool for Fourier analysis, revealing two crucial physical properties. High-fourier components capture intricate textures and fine details, while Low-fourier components represent smoother regions. Approaches like [21, 23, 32] use network structures to handle these components separately for improved detail extraction. However, in denoising, noise mainly resides in high-Fourier components, leading to residual noise in reconstructed images. FFT-transformed Fourier information also offers global statistics, enabling long-range dependency capture. Works such as [24, 34, 36] utilize FFT to enhance neural network representation and generalization. Recent studies [9, 12, 27, 37, 42] split Fourier domain into amplitude and phase, enhancing spatial-Fourier learning. Motivated by these successes, we extensively investigate amplitude and phase components, bridging Fourier and

spatial-spectral domains for high-dimensional HSI denoising, pushing beyond conventional spatial-Fourier domains relationship exploration.

## 3 METHOD

### 3.1 Motivation and Background

The Fourier transform serves as a pivotal component in our method, and a brief review of its principles would be beneficial for comprehending our work. Given a band $x \in \mathbb{R}^{H \times W \times 1}$ in HSI, the Fourier transform of the band $x$ can be formulated as follows:

$$\mathcal{F}(x)(u,v) = \sum_{h=0}^{H-1} \sum_{w=0}^{W-1} x(h,w) e^{-j2\pi\left(\frac{h}{H}u + \frac{w}{W}v\right)}, \quad (1)$$

where $u$ and $v$ are the horizontal and vertical coordinates. $\mathcal{F}(x)$ can be denoted as $\mathcal{F}(x) = R(x) + jI(x)$, where $R(x)$ and $I(x)$ represent the real and imaginary parts of $\mathcal{F}(x)$. Afterward, the amplitude $\mathcal{A}(x)$ and phase $\mathcal{P}(x)$ components from $\mathcal{F}(x)$ can be calculated as:

$$\mathcal{A}(x)(u,v) = \sqrt{R^2(x)(u,v) + I^2(x)(u,v)},$$
$$\mathcal{P}(x)(u,v) = arctan\left(\frac{I(x)(u,v)}{R(x)(u,v)}\right). \quad (2)$$

Correspondingly, the reconstructed process of $R(x)$ and $I(x)$ can be calculated as:

$$\mathcal{R}(x)(u,v) = \mathcal{A}(x)(u,v) \cos \mathcal{P}(x)(u,v),$$
$$\mathcal{I}(x)(u,v) = \mathcal{A}(x)(u,v) \sin \mathcal{P}(x)(u,v). \quad (3)$$

where the Fourier information $\mathcal{F}(x) = R(x) + jI(x)$ can be reconstructed to the original band $x$ by the inverse Fourier transform $\mathcal{F}^{-1}$.

Targeting at HSI denoising, we explore what the $\mathcal{A}$ and $\mathcal{P}$ physically represent in the spatial-spectral domains. As demonstrated and analyzed in Fig. 1 and Fig. 2, we conclude that the noise degradation and photon reflection information predominantly manifest in the amplitude component whereas phase components encapsulate structural information. Therefore, the relationship between the spatial-spectral and the characteristics of the amplitude and phase in the Fourier domain is physically well-defined. Based on the above analysis, we are inspired to restore amplitude and phase components with mutual guidance and explore potential solutions for HSI denoising in both spatial-spectral and Fourier domains. Now we elaborate on our FIDNet, detailed in Fig. 3.

### 3.2 Fourier-prior Integration Denoising Network

*3.2.1 Structure Flow.* For the given noisy sequential input HSIs $Y \in \mathbb{R}^{N \times B \times H \times W}$, we reshape $Y$ as $Y' \in \mathbb{R}^{(N \times B) \times 1 \times H \times W}$ to independently consider spatial and Fourier information, respectively. Then we feed $Y'$ into a $1 \times 1$ convolution layer to generate the initial feature $F_{ini} \in \mathbb{R}^{(N \times B) \times C \times H \times W}$. Subsequently, $F_{ini}$ is fed into both the Spatial Extractor and Fourier Extractor simultaneously to extract distinct spatial and Fourier representations $F_{spa}$ and $F_{fourier}$ with the same shape as $F_{ini}$. Then, the $F_{spa}$ and $F_{fourier}$ in different domains mutually leverage each other's complementary information by a simple but efficient Fourier-Spatial Fusion Module (FSFM) to enhance and merge both representations to get powerful fused feature $F_{bi} \in \mathbb{R}^{(N \times B) \times C \times H \times W}$. Afterward, to incorporate the inherent

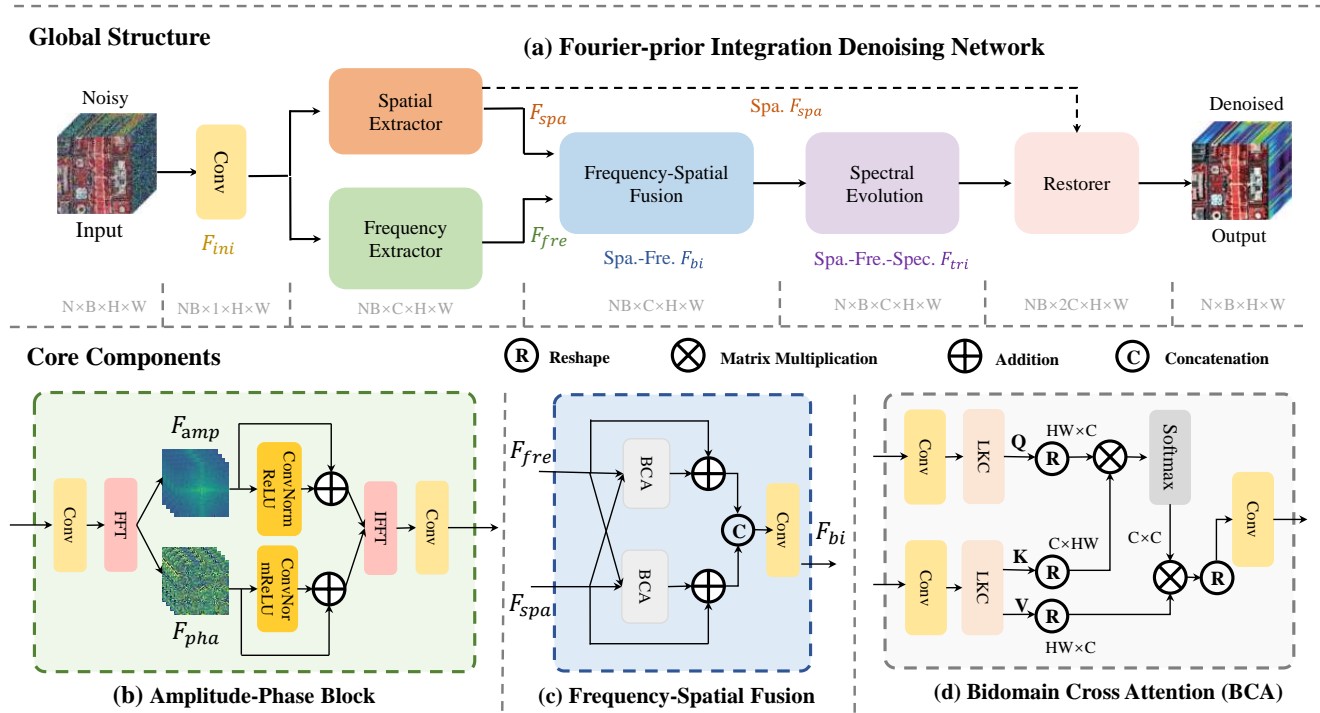

**Figure 3: The overall framework of FIDNet.**

reflective variations and structural consistency constraints of amplitude and phase into the spectral evolution modeling, we cleverly detach the spectral dimension $B$ of the fused features from the batch dimension $(N \times B)$ and transpose it before the feature slices $(H \times W)$. This arrangement maintains the sequence of each slice in the original input image, i.e., $F_{bi}$ is transformed as $F_{bi}^{'} \in \mathbb{R}^{N \times C \times B \times H \times W}$. Since each slice possesses Fourier characteristics, it inherently introduces constraints related to reflective variations and structural consistency. To reduce computational complexity, we directly employ a Vanilla 3-D UNet for spectral evolution learning on the channels $C$ dimension and $F_{tri} \in \mathbb{R}^{N \times 1 \times B \times H \times W}$ is generated. Ultimately, the Restorer reconstructs the clean sequential output HSIs $X_{out} \in \mathbb{R}^{N \times B \times H \times W}$ using $F_{tri}^{'} \in \mathbb{R}^{(N \times B) \times 1 \times H \times W}$ transposed from $F_{tri}$ and the skip connection from $F_{spa}$.

### 3.2.2 Loss Function.
In FIDNet, we propose a joint spatial-spectral and Fourier domain loss for supervising the network training. For the spatial-spectral domain, we minimize the $L_1$ loss of the denoised HSI $X_{out}$ and the ground truth $X_{gt}$, formulated as follows:

$$\mathcal{L}_s = \left\| X_{out} - X_{gt} \right\|_1. \tag{4}$$

In the Fourier domain, we utilize the FFT to transform $X_{out}$ and $X_{gt}$ into the Fourier space, obtaining their corresponding amplitude and phase components. Subsequently, we compute the $L_1$-norms of the amplitude and phase differences between $X_{out}$ and $X_{gt}$ as part of the loss function:

$$\mathcal{L}_{amp} = \left\| \mathcal{A}(X_{out}) - \mathcal{A}(X_{gt}) \right\|_1$$
$$\mathcal{L}_{pha} = \left\| \mathcal{P}(X_{out}) - \mathcal{P}(X_{gt}) \right\|_1. \tag{5}$$

Finally, the overall loss $\mathcal{L}_{total}$ is the combination of $\mathcal{L}_s$, $\mathcal{L}_{amp}$ and $\mathcal{L}_{pha}$, which is formulated as:

$$\mathcal{L}_{total} = \mathcal{L}_s + \lambda(\mathcal{L}_{amp} + \mathcal{L}_{pha}), \tag{6}$$

where $\lambda$ is the weight factor and is empirically set as 0.01.

### 3.2.3 Spatial Extractor & Restorer.
In the Spatial Extractor, it consists of cascaded ConvNormReLU blocks [7] to extract spatial features. The Restorer with a symmetrical structure to reconstruct the clean HSIs.

### 3.2.4 Fourier Extractor & APB..
In Fourier Extractor, it consists of multiple Amplitude-Phase Block (APB, see Fig. 2(b)) to extract distinct features from Fourier domain. In APB, the amplitude and phase components are first calculated by FFT. Then, their features $F_{amp}, F_{pha} \in \mathbb{R}^{(N \times B) \times C \times H \times W}$ are extracted by a cascaded ConvNormReLU block, and reconstructed to Fourier feature by IFFT.

### 3.2.5 FSFM & BCA..
In Fourier-Spatial Fusion Module (FSFM) (see Fig. 2(c)), it adaptively exploits the complementary information from the spatial-Fourier domain by using BCA and generates powerful features for better spectral evolution modeling. Specifically, Bidomain Cross Attention (BCA) (see Fig. 2(d)) utilizes two inputs, the source modality and the complementary modality. To fully interact with these modalities, it generates the query $Q$ from the source

**Table 1: Quantitative evaluation of all the competing methods under mixture noise case on different datasets.**

| Methods | ICVL | | | CAVE | | | PAVIA | | | WDC | | |
|---|---|---|---|---|---|---|---|---|---|---|---|---|
| | PSNR | SSIM | SAM | PSNR | SSIM | SAM | PSNR | SSIM | SAM | PSNR | SSIM | SAM |
| Noisy | 13.97 | 0.3392 | 0.8987 | 14.17 | 0.4188 | 1.1371 | 13.89 | 0.3400 | 0.9746 | 13.98 | 0.2148 | 1.0120 |
| LRTDTV [28] | 34.46 | 0.9184 | 0.1127 | 33.82 | 0.9085 | 0.2938 | 29.65 | 0.8963 | 0.2445 | 36.33 | 0.8597 | 0.2148 |
| LLRGTV [11] | 31.39 | 0.8756 | 0.2538 | 27.12 | 0.7221 | 0.6567 | 28.41 | 0.8923 | 0.3142 | 34.32 | 0.8260 | 0.3150 |
| QRNN3D [30] | 39.22 | 0.9904 | 0.0809 | 36.55 | 0.9825 | 0.4244 | 32.93 | 0.9698 | 0.1570 | 33.96 | 0.8744 | 0.1344 |
| GRNet [2] | 31.67 | 0.9557 | 0.1431 | 28.44 | 0.8899 | 0.6329 | 26.57 | 0.8536 | 0.2815 | 25.14 | 0.7682 | 0.3255 |
| MAC-Net [33] | 30.75 | 0.9332 | 0.2673 | 28.53 | 0.8920 | 0.6234 | 27.34 | 0.8813 | 0.3530 | 30.74 | 0.7740 | 0.5371 |
| T3SC [1] | 35.68 | 0.9790 | 0.1389 | 33.61 | 0.9728 | 0.4137 | 31.39 | 0.9523 | 0.2314 | 30.49 | 0.9064 | 0.1972 |
| MAN [13] | 35.85 | 0.9691 | 0.1458 | 35.15 | 0.9712 | 0.4990 | 32.94 | 0.9711 | 0.1820 | 32.77 | 0.8065 | 0.1638 |
| SST [16] | 39.58 | 0.9928 | 0.0480 | 34.89 | 0.9616 | 0.4095 | 32.85 | 0.9588 | 0.1555 | 30.59 | 0.8924 | 0.1860 |
| SERT [17] | 40.44 | 0.9941 | 0.0470 | 35.86 | 0.9737 | 0.3403 | 33.28 | 0.9653 | 0.1451 | 31.33 | 0.9098 | 0.1609 |
| HSDT [14] | 40.76 | 0.9940 | 0.0505 | **38.52** | **0.9892** | 0.2362 | 34.25 | **0.9768** | 0.1295 | 37.79 | 0.9465 | 0.1049 |
| FIDNet (Ours) | **40.89** | **0.9941** | **0.0463** | 37.89 | 0.9888 | **0.2219** | **34.26** | 0.9722 | **0.1266** | **39.36** | **0.9523** | **0.1042** |

**Table 2: Quantitative evaluation of all the competing methods under different complex noise cases on the KAIST dataset.**

| Methods | Non-i.i.d Gaussian | | | Gaussian+Stripe | | | Gaussian+Deadline | | | Gaussian+Impulse | | | Gaussian+Mixture | | |
|---|---|---|---|---|---|---|---|---|---|---|---|---|---|---|---|
| | PSNR | SSIM | SAM | PSNR | SSIM | SAM | PSNR | SSIM | SAM | PSNR | SSIM | SAM | PSNR | SSIM | SAM |
| Noisy | 18.26 | 0.6145 | 1.0741 | 18.10 | 0.6136 | 1.0746 | 17.23 | 0.5770 | 1.0960 | 15.00 | 0.4625 | 1.0965 | 13.70 | 0.4216 | 1.1133 |
| LRTDTV | 37.33 | 0.9510 | 0.2195 | 37.16 | 0.9498 | 0.2113 | 36.49 | 0.9413 | 0.2354 | 37.03 | 0.9505 | 0.2445 | 35.60 | 0.9355 | 0.2503 |
| LLRGTV | 36.66 | 0.9109 | 0.3678 | 36.68 | 0.9152 | 0.3510 | 34.76 | 0.8761 | 0.4268 | 33.61 | 0.8502 | 0.5109 | 32.14 | 0.8166 | 0.5218 |
| QRNN3D | 39.42 | 0.9874 | 0.2085 | 39.24 | 0.9877 | 0.2086 | 38.99 | 0.9868 | 0.2167 | 38.15 | 0.9853 | 0.2505 | 37.10 | 0.9831 | 0.2934 |
| GRNet | 30.58 | 0.9131 | 0.5235 | 30.41 | 0.9093 | 0.5298 | 30.22 | 0.9044 | 0.5280 | 28.77 | 0.8870 | 0.5996 | 28.29 | 0.8847 | 0.5780 |
| MAC-Net | 34.33 | 0.9537 | 0.4795 | 33.37 | 0.9520 | 0.4733 | 32.82 | 0.9428 | 0.4718 | 31.02 | 0.9205 | 0.5925 | 28.41 | 0.9041 | 0.5840 |
| T3SC | 37.79 | 0.9915 | 0.1603 | 37.68 | 0.9915 | 0.1678 | 37.04 | 0.9903 | 0.1888 | 34.87 | 0.9808 | 0.2561 | 33.62 | 0.9750 | 0.2958 |
| MAN [13] | 38.76 | 0.9887 | 0.2076 | 38.54 | 0.9889 | 0.2102 | 38.16 | 0.9881 | 0.2117 | 36.38 | 0.9779 | 0.3220 | 35.27 | 0.9704 | 0.3818 |
| SST [16] | 36.79 | 0.9774 | 0.2267 | 36.59 | 0.9779 | 0.2294 | 36.34 | 0.9762 | 0.2323 | 35.62 | 0.9726 | 0.2622 | 34.65 | 0.9638 | 0.3050 |
| SERT [17] | 37.77 | 0.9851 | 0.1918 | 37.57 | 0.9849 | 0.1956 | 37.35 | 0.9836 | 0.1976 | 36.58 | 0.9803 | 0.2262 | 35.71 | 0.9761 | 0.2459 |
| HSDT [14] | 40.18 | 0.9905 | 0.1413 | 40.07 | 0.9909 | 0.1406 | 39.76 | 0.9920 | 0.1465 | 39.47 | 0.9913 | 0.1537 | 38.31 | 0.9904 | 0.1582 |
| FIDNet | **40.52** | **0.9935** | **0.1236** | **40.30** | **0.9937** | **0.1257** | **40.14** | **0.9929** | **0.1300** | **39.55** | **0.9930** | **0.1381** | **38.84** | **0.9907** | **0.1489** |

modality and obtains the key $K$ and value $V$ from complementary modality by applying different large kernel convolutional (LKC) layers [8], providing the large modality receptive field. After that, the attention process of the BCA can be written as follows:

$$\text{Attention}(\mathbf{Q}, \mathbf{K}, \mathbf{V}) = \text{Softmax}\left(\frac{\mathbf{Q}\mathbf{K}^T}{\sqrt{d_k}}\right)\mathbf{V}, \qquad (7)$$

where $\sqrt{d_k}$ is a scalar as defined in [4]. To capture the incorporation of modalities along the channel dimension, multiplication is calculated between the corresponding elements along the channel dimension.

## 4  EXPERIMENTS

In this section, we comprehensively assess the performance of various methods on synthetic and real datasets and substantiate the efficacy of our proposed approach through a combination of quantitative and qualitative analyses.

### 4.1  Benckmarks

*4.1.1  Datasets.* Following the experimental setting in [17], we select 100 images from the ICVL[1] dataset for training. For testing of synthetic experiments, we randomly select 50 images from ICVL (excluding the training set), 10 images from the KAIST [3] and 30 images from CAVE [22]. In addition, the remote sensing datasets WDC Mall[2] and Pavia Center [6] are adopt further verify the generalization capabilities of each model. For the real experiments, the HYDICE Urban [19] and AVIRIS Indian Pines [15] are adopted for testing.

*4.1.2  Training data.* In this section, we detail our dataset selection and preprocessing. From the *ICVL* dataset, containing a total of 201 images, we randomly choose 100 images with the size of 1392 × 1300 × 31 for training. Specifically, these selected images are firstly center-cropped into a size of 1024 × 1024 × 31. To diversify the training set, we apply augmentations such as random flipping,

---

[1]http://icvl.cs.bgu.ac.il/hyperspectral/

[2]https://engineering.purdue.edu/~biehl/MultiSpec/hyperspectral.html

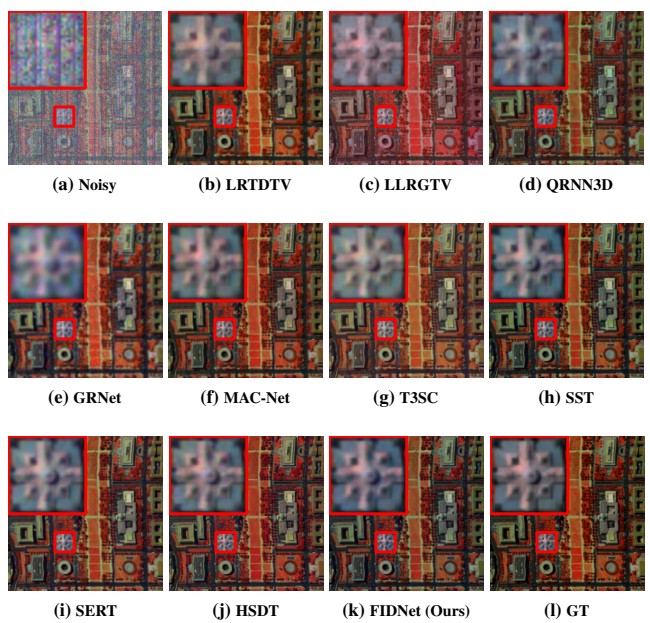

**Figure 4: Denoising visual comparison of the WDC Mall dataset. The visual image is synthesized by HSI bands 76, 43, and 10.**

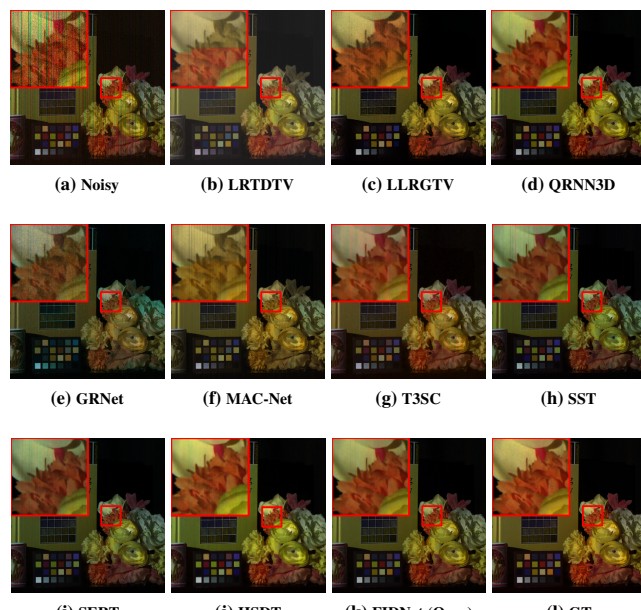

**Figure 5: Denoising visual comparison of the KAIST dataset. The visual image is synthesized by HSI bands 29, 19, 9.**

**Table 3: The number and size of the testing set for all datasets.**

| Datasets | Number | Original size | Cropped size |
|---|---|---|---|
| ICVL | 50 | $1392 \times 1300 \times 31$ | $512 \times 512 \times 31$ |
| CAVE | 30 | $512 \times 512 \times 31$ | $512 \times 512 \times 31$ |
| KAIST | 10 | $3376 \times 2704 \times 31$ | $2048 \times 2048 \times 31$ |
| Pavia Center | 1 | $1096 \times 715 \times 102$ | $384 \times 384 \times 102$ |
| WDC Mall | 1 | $1208 \times 307 \times 191$ | $256 \times 256 \times 191$ |
| Indian Pines | 1 | $145 \times 145 \times 220$ | $144 \times 144 \times 220$ |
| Urban | 1 | $307 \times 307 \times 210$ | $224 \times 224 \times 210$ |

rotation, and scaling. Consequently, we create cube data of size 64 × 64 × 31 at scales 1:1, 1:2, and 1:4, with strides 64, 32, and 32, resulting in 53,000 training samples.

*4.1.3 Testing data.* Regarding the testing set, we consider both synthetic and real-world data. The details of testing datasets are shown in Table 3. For synthetic experment, the *ICVL*, *CAVE* [22], *KAIST* [3], *Pavia Center* [6] and *WDC Mall* are employed as testing data. For real-world data, we adopt the *Indian Pines* [15] and *Urban* [19] contained unknown noises as testing data.

*4.1.4 Methods and Metrics.* To demonstrate the effectiveness of our approach, we evaluate its performance against several state-of-the-art methods, including two traditional models (LRTDTV [28] and LLRGTV [11]) and five deep-learning methods (QRNN3D [30], GRNet [2], MACNet [33], T3SC [1], MAN [13], SST [16], SERT [17], HSDT [14]). We utilize widely accepted image quality metrics, including PSNR, SSIM [29], and SAM [39], for a comprehensive assessment of denoising performance.

## 4.2 Implementation Details

To simulate complex noisy scenarios in real-world HSIs, we employ five noise patterns: Non-i.i.d Gaussian noise, Non-i.i.d Gaussian noise with stripe noise, deadline noise, impulse noise, and a mixture of these patterns. Comprehensive noise configuration specifics can be referenced in [17]. We train FIDNet for 50 epochs with a batch size of 16 and the learning rate is initialized as $2 \times 10^{-4}$ and decayed to $1 \times 10^{-4}$ after 40 epochs with Adam optimizer, where the $\beta_1$ and $\beta_2$ are set to 0.9 and 0.99 respectively. We set the basic channel $C$ = 12. The traditional models are run on the Intel Core i7-13700KF CPU, whereas all deep-learning techniques are trained on NVIDIA GeForce RTX 3090 GPUs.

## 4.3 Experimental Results and Analysis

In this section, we present additional visual results on the ICVL and CAVE test sets to further illustrate the performance of FIDNet. It is evident from Fig. 6 of ICVL that noticeable stripe noise remains in SERT's result, whereas FIDNet successfully removes the majority of the noise, resulting in superior visual quality. From Fig 6 of CAVE, FIDNet exhibits colors and textures that are much closer to the ground truth, highlighting its superior performance.

In synthetic and real-world experiments, all DL methods are trained on the ICVL dataset. For methods (GRNet, T3SC, SST, SERT) that are not flexible with large spectral numbers, we employ a sliding window manner for denoising and take the average value as the final result.

*4.3.1 Synthetic Experiments.* To underscore FIDNet's robustness, we initially assess its performance on four diverse benchmarks: ICVL, CAVE, and remotely sensed datasets Pavia Center and WDC

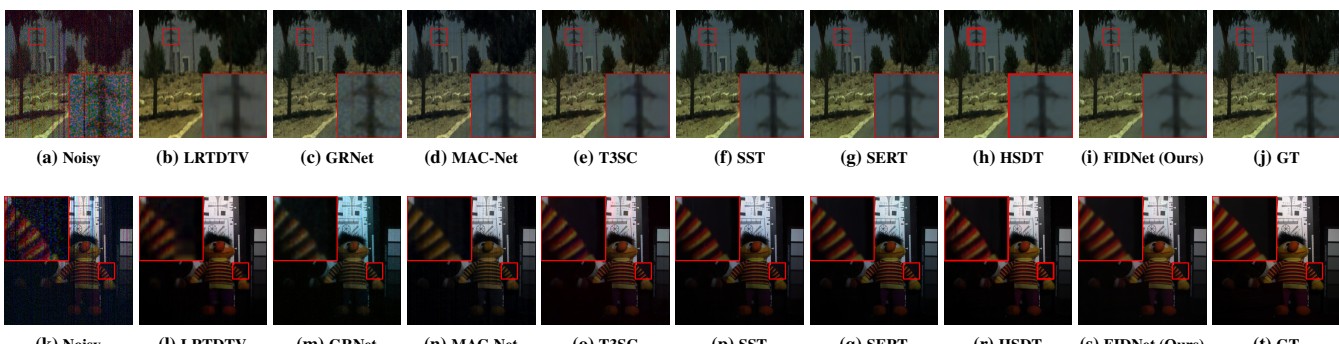

**Figure 6: Denoising visual comparison of ICVL and CAVE dataset. The visual image is synthesized by HSI bands 29, 19, 9.**

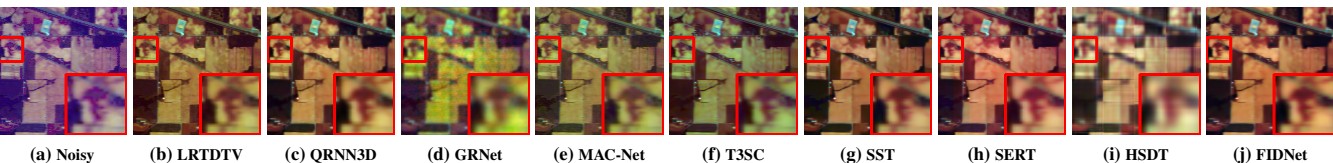

**Figure 7: Visual comparison of the real-world dataset Indian Pines. The visual image is synthesized by HSI bands 127, 24, 2.**

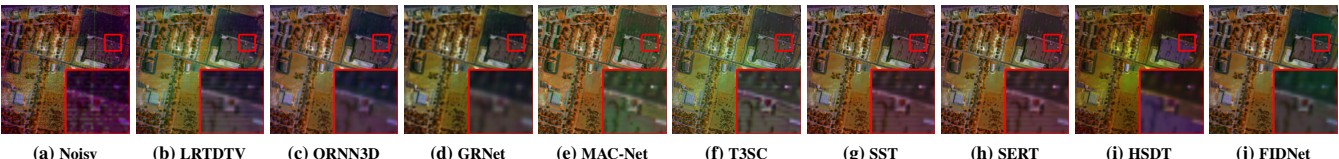

**Figure 8: Visual comparison of the real-world dataset Urban. The visual image is synthesized by HSI bands 102, 138, 202.**

Mall, incorporating intricate mixture noise scenarios. As depicted in Table 1, SOTA methods exhibit remarkable results on ICVL but reveal limitations in generalization across the other datasets. In contrast, FIDNet consistently excels, notably surpassing SST and SERT by **8.77 dB** and **8.03 dB** in PSNR on WDC Mall, possibly due to overfitting within specific data domains by other methods. Moreover, visual representation using the WDC Mall dataset in Fig. 4 illustrates our method's efficacy, preserving image structure while minimizing residual noise. Additionally, we thoroughly evaluate FIDNet's denoising performance on the KAIST dataset, encompassing various complex noise scenarios and a substantial spatial resolution. Quantitative and qualitative evaluations, as showcased in Table 2 and Fig. 5, respectively, establish FIDNet's superiority over SOTA approaches in all metrics. Furthermore, performance disparities between KAIST and ICVL training datasets expose deficiencies in recent SOTA methods, SST, SERT, and HSDT, emphasizing FIDNet's robust generalization and dataset insensitivity. This success can be attributed to FIDNet's adept utilization of Fourier prior knowledge, effectively decoupling noise and structure based on physical principles, and enhancing its capacity to capture spectral variations while preserving structural coherence within the spectral domain.

**Table 4: Comparison on Parameter count (M), FLOPS (G), and runtime (sec.). These metrics are compared using the ICVL dataset with a size of 512×512×31. The runtime of each method is tested on the NVIDIA 3090 GPU.**

| Methods | GRNet | MACNet | T3SC | SST | SERT | HSDT | FIDNet |
|---------|-------|--------|------|------|------|------|--------|
| PSNR    | 31.67 | 30.75  | 35.68| 39.58| 40.44| 40.76| 40.89  |
| Params  | 41.44 | 0.43   | 0.83 | 4.10 | 1.91 | 0.13 | 0.48   |
| GFLOPS  | 610.7 | -      | -    | 2082.4| 1018.9| -   | 2115.6 |
| Times   | 0.466 | 2.709  | 0.758| 2.265| 0.872| 1.032| 0.841  |

*4.3.2 Real-World Experiments.* To address real-world noise and produce visually appealing HSIs, we conduct denoising experiments on real-world Indian Pines and Urban datasets. The visual outcomes, depicted in Fig.7 and Fig.8, highlight FIDNet's consistent production of realistic, artifact-free textures. This robust performance substantiates the efficacy of our approach.

*4.3.3 Efficiency Experiments.* The practical applicability of a model hinges on its efficiency, encompassing factors like parameter counts (Params, M), computational complexity (FLOPS, G), and

runtime (s). Therefore, we compare the efficiency of each method on ICVL datasets. The results in Table 4 indicate that the simple yet efficient FIDNet achieves low parameter costs and efficient inference times.

## 4.4 Ablation Study

*4.4.1 Network Structure.* The global structure of FIDNet consists of four main modules: Spatial Extractor, Fourier Extractor, Spectral Evolution Module, and Restorer. The Spatial Extractor and Restorer are symmetric, each composed of two cascaded ConvNorm-ReLU blocks [7]. The Fourier Extractor consists of two proposed Amplitude-Phase Blocks (APBs). The Spectral Evolution module is a 3-D UNet composed of Residual Dense Blocks (RDBs) [40].

*4.4.2 Components and Losses.* In FIDNet, the Fourier prior is derived through the Fourier Extractor, while the Fourier-Spatial Fusion Module (FSFM) seamlessly combines Fourier and spatial features, accompanied by the domain-specific loss function. To validate the effectiveness of this proposed Fourier prior and its associated losses, we establish a baseline with a Spatial Extractor, Spectral Evolution, and Restorer, and enhance it with the Fourier Extractor, denoted as baseline+F. This involves direct element-wise addition of spatial-Fourier domain features. These models are assessed using loss functions $L_s$ in Eq.4 and $L_{total}$ in Eq.6, and their performance and efficiency are compared in Table 5.

**Table 5: Ablation study on the effect of Fourier Extractor and FSFM using different loss functions under WDC Mall dataset.**

| Models | PSNR ($L_s$) | PSNR ($L_{total}$) | Params (M) | GFLOPS |
|---|---|---|---|---|
| Baseline | 38.39 | 38.53 | 0.46 | 1470.3 |
| Baseline+F | 38.74 | 39.02 | 0.47 | 1545.0 |
| FIDNet | 39.03 | 39.36 | 0.48 | 1641.1 |

Key insights drawn from the results are as follows: Integrating Fourier priors substantially improves the baseline model's denoising ability. However, due to distinct feature expressions in spatial and Fourier domains, FSFM is introduced for seamless fusion, leading to further performance enhancement. Furthermore, supervised training using $L_{total}$ on amplitude and phase components boosts performance across all models, underscoring its efficacy.

## 4.5 Number of Feature Channel

To investigate the influence of module channel numbers, experiments are conducted on the WDC Mall dataset, as depicted in Table 6. Unlike the Spectral Evolution Module, the Spatial Extractor, Fourier Extractor, and Restorer focus on feature extraction from individual spectral bands. Therefore, we maintain consistency in the feature channel numbers $C_{SFR}$ for the Spatial Extractor, Fourier Extractor, and Restorer.

The empirical findings underscore that augmenting parameter counts and computational costs may yield incremental performance enhancements (Model 6). However, it is imperative to weigh this against the associated trade-off between resource expenditure and performance gains, which may not always be fully justified. In contrast, a strategic choice of 12 channels for both $C_{SFR}$ and the channel

**Table 6: Different channel combinations between modules. $C_{SFR}$ and $C_{Spec}$ denote the channel number of (Spatial Extractor, Fourier Extractor, Restorer), and the Spectral Evolution Module, respectively.**

| Models | $C_{SFR}$ | $C_{Spec}$ | PSNR | Parmas | GFlops |
|---|---|---|---|---|---|
| 1 | 8 | 8 | 37.88 | 0.22 | 820.4 |
| 2 | 8 | 12 | 38.52 | 0.47 | 1500.8 |
| 3 | 8 | 16 | 38.56 | 0.82 | 2444.4 |
| 4 | 12 | 8 | 38.44 | 0.25 | 1138.3 |
| 5 | 12 | 12 | 39.36 | 0.48 | 1629.5 |
| 6 | 12 | 16 | 39.47 | 0.85 | 2784.7 |
| 7 | 16 | 8 | 38.53 | 0.29 | 1570.1 |
| 8 | 16 | 12 | 39.24 | 0.54 | 2272.9 |
| 9 | 16 | 16 | 39.27 | 0.89 | 3239.0 |

number of the Spectral Evolution Module $C_{Spec}$ (Model 5) strikes a harmonious equilibrium between performance optimization and computational efficiency, underscoring the importance of meticulous parameter selection in achieving optimal model efficacy.

## 4.6 Weight of Fourier Loss

To enhance the acquisition of Fourier features during training, we integrate Fourier loss into the training process. By adjusting the weights $\lambda$ assigned to the Fourier loss, we observe diverse learning effects. As illustrated in Table 7, the network achieves peak performance when the value of $\lambda$ is configured to 0.01.

**Table 7: Evaluation results on WDC Mall using different values of weights $\lambda$ for Fourier loss.**

| Models | $\lambda$ | PSNR | SSIM | SAM |
|---|---|---|---|---|
| 1 | 1 | 38.62 | 0.9440 | 0.1121 |
| 2 | 0.1 | 38.71 | 0.9435 | 0.1269 |
| 3 | 0.01 | 39.36 | 0.9523 | 0.1078 |
| 4 | 0.001 | 39.28 | 0.9490 | 0.1100 |
| 5 | 0 | 39.03 | 0.9484 | 0.1087 |

## 5 CONCLUSION

This paper explores the attributes of amplitude and phase components in the spatial-spectral domain for HSI denoising. Amplitude encapsulates noise and photon reflections, while the phase encodes structural information. Leveraging these insights, we introduce FID-Net, a novel solution seamlessly integrating Fourier priors into the spatial-spectral context. FIDNet adeptly fuses spatial and Fourier features, harnessing their complementary strengths to yield advantageous attributes from both domains for better spectral modeling, outperforming mainstream denoising methods in comprehensive experiments. It establishes a new benchmark on datasets, showcasing versatility in handling varying spectral bands for real-world applicability.

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
