# OpenReview forum: "Bridging Fourier and Spatial-Spectral Domains for Hyperspectral Image Denoising"
_acmmm.org/ACMMM/2024/Conference — MM2024 Poster_

### Official Review · Reviewer_C64F · 2024-05-16

**Rating:** 4
**Confidence:** 4

**Summary:**

In hyperspectral image (HSI) denoising, existing methods often overlook the untapped potential in the Fourier domain, focusing predominantly on the spatial-spectral domain. This paper introduces a novel approach, the Fourier-prior Integration Denoising Network (FIDNet), which leverages insights from the Fourier domain to synergistically interact with spatial-spectral domains, resulting in good HSI denoising performance demonstrated across synthetic and real-world benchmark datasets.

**Strengths:**

address HSI denoising by bridging the information from the Fourier and spatial-spectral domains.

**Limitations:**

There is no discernible improvement in visual results attributed to the incorporation of Fourier domain feature modeling, as evidenced by Figures 4 through 8.

The rationale behind separately extracting spatial and spectral features from the projected HSI cube, followed by fusion, lacks clear justification. It begs the question of why not directly model the spatial-spectral frequency features?

Comparative analysis of computational parameters and gigaflops across all methods, particularly the transformer-based approaches, is essential for a comprehensive evaluation, given that this paper is founded on multi-head self-attention.

**Suitability:**

2

---

### Official Review · Reviewer_Bx2u · 2024-05-24

**Rating:** 4
**Confidence:** 3

**Summary:**

This paper proposes a neural network that interacts between the spatial-spectral domain and the frequency domain. By extracting information from both domains through two separate network branches and fusing them, the interaction between the spatial-spectral and frequency domains is achieved. Additionally, the article studies the physical significance of the Fourier amplitude spectrum and phase spectrum through an experiment of exchanging amplitude spectrum. The results show that this method surpasses existing state-of-the-art methods.

**Strengths:**

1. This paper studies the correlation between different bands of hyperspectral images through Fourier transform, effectively demonstrating that the differences between various spectral bands can be reflected in the differences in the frequency spectrum, which is interesting.
2. This paper combines Fourier frequency domain learning with cross attention to design an efficient network structure.
3. The proposed method achieves comparable results with the existing state-of-the-art.

**Limitations:**

1. The application of frequency domain learning and cross attention to the task of hyperspectral image denoising somewhat lacks novelty, as they have been extensively studied in image processing.
2. The properties of the frequency spectrum and phase spectrum in Fourier domain have been extensively studied. Moreover, Fourier transform can not effectively separate signal from noise, as can be seen in Figure 2. The paper makes some overstatements about the separation of noise by the Fourier transformation.
3. The frequency domain information and spatial domain information are not separate and can be explicitly transformed into each other. Whether it is meaningful to use the attention mechanism to explore the correlation of signals with clear relationships is not well investigated.
4. Although the experiments claim to follow [17], the experimental results of [17] in Table 1 and Table 2 differ from those reported in the original paper. Since the experiments in [17] are easy to reproduce, the reviewer is curious about what factors led to this discrepancy.

**Suitability:**

3

---

### Official Review · Reviewer_2iZN · 2024-06-05

**Rating:** 4
**Confidence:** 3

**Summary:**

The paper introduces a new denoising model for hyperspectral images (HSI) called the Fourier-prior Integration Denoising Network (FIDNet). FIDNet combines information from the fourier and spatial domains by independently extracting spatial and fourier features through a dual-branch structure and integrating these representations to facilitate the exchange of complementary information. Experiments validate the effectiveness of the proposed method.

**Strengths:**

1. The authors effectively reveal the roles of amplitude and phase components in spatial degradation and spectral evolution through visualizations in Fig. 2, thus justifying the introduction of Fourier priors. The idea is both reasonable and practical.
2. FIDNet is designed to utilize information from both spatial and spectral domains complementarily, which significantly enhances HSI denoising performance.
3. The proposed frequency extractor exploits the characteristics of amplitude-phase components. The frequency-spatial fusion module, employing cross-attention, is both simple and effective. The ablation studies demonstrate the effectiveness of the two components.
4. Experiments on both synthetic and real-world benchmark datasets demonstrate the effectiveness of the proposed method.

**Limitations:**

1. The introduction of the proposed method is unclear. In Fig. 3, the spectral evolution is mentioned without adequate introduction or referencing previous methods; the specific references are found in L220 and L825 but do not point to the same approach, which is puzzling.
2. Compared to the latest method, HSDT, the performance improvements of the proposed method are not significant on some datasets (ICVL) and even lower on others (CAVE, PAVIA). This necessitates further validation of the proposed method's superiority. Additionally, visual comparisons in Figs. 4 and 5 do not show a significant difference over HSDT.
3. The spectral evolution applies the Vanilla 3D UNet. Why not use the state-of-the-art HSDT? It would be interesting to explore if using HSDT, with similar complexity and parameters, could surpass HSDT performance.
4. Detail issue: In Secs. 3.2.4 and 3.2.5, there is an error in the figure references; it should be Fig. 3 instead of Fig. 2.

**Suitability:**

3

---

### Meta-Review · Area_Chair_Ni7w · 2024-07-02

**Recommendation:** Accept (Poster)
**Confidence:** 5

**Metareview:**

The paper presents a potentially impactful idea with well-conducted experiments, but it struggles to clearly outperform existing methods and to convincingly justify all of its design choices. The authors are advised to address these issues in future work or revisions by clarifying the unique contributions of their method, enhancing the comparative analysis, and providing clearer justifications for their design decisions.